# Coordination Characteristics Analysis of Deformation between Polymer Anti-Seepage Wall and Earth Dam under Traffic Load

Hongyuan Fang [1], Hong Zhang [1], Binghan Xue [1,*], Jianglin Gao [2], Yan Li [2], Xinlei Gao [3] and Aiping Tian [4]

1   School of Water Conservancy Engineering, Zhengzhou University, Zhengzhou 450000, China; fanghongyuan1982@163.com (H.F.); hongzhang0053@163.com (H.Z.)
2   Jiangxi Academy of Water Science and Engineering, Nanchang 330000, China; gaojianglin@163.com (J.G.); liyan6609@163.com (Y.L.)
3   Harbin Institute of Technology National Engineering Research Center of Urban Water Resources, Harbin 150006, China; lgeg2018@163.com
4   Shanghai Dike and Pump Gate Construction and Operation Centre, Shanghai 200000, China; tianfxq@163.com
*   Correspondence: xuebinghan@zzu.edu.cn

**Abstract:** Polymer anti-seepage walls have been widely used in the anti-seepage reinforcement projects of earth dams. Dam crest is always supposed to meet the requirements of traffic load which has significant influence on the dam body and anti-seepage wall. In order to analyze the coordination characteristics of the deformation between polymer anti-seepage wall and dam under traffic loads, a 3D finite element model of an earth dam that considers the coupling effect of seepage field and stress field was established. Besides, the influence of load amplitude, vehicle speed and driving position on the stress and deformation characteristics of polymer anti-seepage wall and dam was analyzed, with the displacement difference between dam and wall, wall Mises stress and wall subsidence as indicators. The results show that, compared with vehicle speed, the load amplitude and vehicle speed of traffic load exerted a greater impact on the coordination characteristics of the deformation of the dam. The variation range of the displacement difference caused by axial load change reached 87.1%, while that resulted from driving position change reached 90.3%. That is, when the passing vehicle has a light axle load and passes quickly over the anti-seepage wall, it has less impact on the dam.

**Keywords:** earth dam; polymer anti-seepage wall; traffic load; deformation coordination

## 1. Introduction

Dams bring huge economic, social and environmental benefits in flood control, water supply, irrigation, shipping and other aspects, playing an important role in economic and social progress. Seepage is one of the key issues of dam safety, and building anti-seepage walls are one of the common measures for dam seepage prevention [1]. With the appearance of various emerging materials, the materials of anti-seepage walls have also been improved, which are developed from the original ordinary concrete to reinforced concrete, plastic concrete, clay concrete, polymer anti-seepage walls, and so on. Non-aqueous reactive polyurethane, as an anti-seepage reinforcement material for dams, is featured with light weight, large expansion force, non-pollution, good anti-seepage performance, excellent durability and mechanical properties, etc. Compared with rigid concrete anti-seepage walls with large elasticity modulus, polymer anti-seepage walls make up for these deficiencies [2,3].

At present, some studies have been conducted on the properties of polymer grouting materials, grouting technology and construction effects [3–11], achieving significant progress and innovation in the construction technology of polymer anti-seepage walls. Li Jia et al. [12–14] studied the seismic response characteristics of earth dam structure of polymer anti-seepage walls by means of a physical centrifuge model, and compared the seismic response characteristics of earth dams with polymer anti-seepage walls and concrete core walls. The non-aqueous reactive two-component polyurethane polymer

grouting material used in the technology of the polymer anti-seepage wall of the earth dam belongs to polyurethane materials. Although the anti-seepage wall exerts significant anti-seepage effect on the dam, the material interface under load is often the "weak link" of the structure [15,16] due to the different material properties between the soil body and the anti-seepage wall. Additionally, some progresses have been made on the study of material interface in recent years [17–19]. Deformation difference between the two material regions will inevitably occur during the construction period and water storage period. The interface between traditional concrete anti-seepage wall and dam often has certain defects and is not an ideal bonding state. As a result, such deformation differences may cause cracks in the material interface, the function of the dam is seriously affected. Therefore, it is necessary to study the coordination characteristics of the dam deformation. The evaluation indicators that can be used as deformation coordination properties include major principal stress, wall stress level, the maximum subsidence of dam, the change rate of wall subsidence, as well as vibration compaction time [20].

The dam crest is always supposed to meet the requirements of traffic load, and the traffic load exerts great influence on the dam body and leads to irreversible deformation of the dam body. With the increase of traffic flow and load capacity, the traffic load gradually has greater impact on the dam body. Sarkar examined the flexible pavement dynamic response under single, tandem and tridem axles at different speeds. Using two different hot-mix asphalt (HMA) layer thicknesses, the dynamic effects of moving axles were investigated on critical responses. These responses include the tensile strain at the bottom of asphalt layer, compressive strain on the top of subgrade and tensile and compressive strain on the surface layer [21]. Zhao et al. presented a numerical investigation into the plastic/dynamic characteristics of a saturated porous medium (capping and subgrade) subjected to moving axle loads. A detailed numerical analysis is described, whereby a coupled fluid-dynamic framework is developed for the saturated porous medium in conjunction with a generalized plasticity model, in order to examine the cyclic loading response of a soft subgrade soil. A relationship between the train speed, track settlement and drainage capacity of a sub-ballast (capping) layer is established [22]. Guo et al. comprehensively analyzed the dynamic response of full-depth asphalt pavement under moving load, a three-dimensional model of pavement structure and dynamic load moving zone are established based on ABAQUS finite element software. Based on the time history curves of different structures, the stress–strain states at the bottom of each structural layer in different structures under moving load are analyzed [23]. Liu et al. conducted hollow torsional shear tests on fiber-reinforced aeolian soil involving varying fiber contents, cyclic deviator stresses, cyclic shear stresses and consolidation confining pressures using the Small-Strain Hollow Cylinder Apparatus. This enabled an investigation of the deformation characteristics and noncoaxial angle changes of fiber-reinforced aeolian soil under a heart-shaped stress path [24]. Qian et al. performed a series of cyclic torsional shear tests to investigate the effect of principal stress rotation (PSR) on the stress–strain behaviors of saturated soft clay. The traffic-load-induced shear stress path was used in the cyclic test and the investigation mainly concerned the influence of PSR on the shear stiffness and non-coaxiality [25]. Currently, the analysis on traffic load mainly focuses on the field of roads, while few studies are conducted on the impact of traffic load on dams with polymer anti-seepage walls. Therefore, analyzing the influence of traffic load on dam safety is of important theoretical and practical value.

In order to analyze the stress and deformation characteristics of polymer anti-seepage walls and dams under traffic load, a three-dimensional finite element model of polymer anti-seepage dam wall that considers the coupling effect of seepage field and stress field was established, the influence of axle load, vehicle speed and driving position on the coordination characteristics of dam deformation was simulated, and the law of stress and deformation of dam under traffic load was revealed, which provides a basis for the design and safety evaluation of polymer anti-seepage walls in dam reinforcement engineering.

The remaining sections are arranged as follows: the indexes for evaluating the deformation coordination characteristics are introduced in Section 2; an overview of the model

and load settings is presented Section 3; the calculation results are analyzed in Section 4; and the summary and conclusion are made in Section 5.

## 2. Deformation Coordination Index

Uneven subsidence will occur in the interface of the dam body and the anti-seepage wall due to the inconsistency of mechanical properties. The cross section of the dam body will dislocate in a nearly vertical direction because of the uneven subsidence. Meanwhile, the development of the uneven longitudinal displacement of the dam body will cause the dam body to separate on both sides of the failure surface, thus forming cracks with a certain width. In terms of the general dam body, the failure surface caused by the subsidence difference shows cracks of different widths. Thus, some indexes for evaluating the coordination characteristics of the deformation were proposed [26], thus that the judgment and analysis on the deformation difference of the dam could be carried out. The displacement difference was selected as the index to evaluate the deformation coordination of the polymer anti-seepage wall and the dam, and its definition is as follows:

The formula for the vertical displacement difference between the anti-seepage wall and the dam body is as follows:

$$H_y = D_{xi}\big|_j - d_{xi}\big|_j \tag{1}$$

where $d_{xi}\big|_j$ means the vertical displacement of point $i$ of the anti-seepage wall at time $j$ and $D_{xi}\big|_j$ refers to the vertical displacement of the corresponding point $i$ of the dam body at time $j$.

The formula for the horizontal displacement difference between the anti-seepage wall and the dam body is as follows:

$$H_x = D_{fi}\big|_j - d_{fi}\big|_j \tag{2}$$

where $d_{fi}\big|_j$ indicates the deflection of point $i$ of the anti-seepage wall at time $j$ and $D_{fi}\big|_j$ stands for the lateral displacement of the corresponding dam body $i$ at time $j$.

## 3. Calculation Model
### 3.1. Overview of Model

According to the information on the Yellow River dam section which is crossed by the Zhengjiao inter-city railroad bridge, a 3D finite element model as shown in Figure 1 was established by using ABAQUS software, in which the height of the dam body is 15.0 m, and that of the dam foundation is 10.0 m, the overall length along the *X*-axis is 124.5 m and the width along the *Z*-axis direction is 2.1 m. The height of the anti-seepage wall is 17.0 m, and its lower part is embedded in the dam foundation for 2.0 m. The C3D8P element with pore pressure degree of freedom was used for the solution and the Mohr–Coulomb elastic-plastic constitutive model was adopted for the soil, assuming that the foundation is homogeneous, the literatures [27,28] carried out studies on the randomness of parameters, and the linear elastic constitutive model was employed for the wall. The total number of elements is 30,024 and that of nodes is 35,087, in which the number of the units and nodes of the anti-seepage wall are 1584 and 2210, respectively. The material parameters are shown in Table 1. Traffic load was applied at the top of the dam, and the seepage boundary was set within the range from the bottom of the dam near water to the height of 13.5 m. Goodman element without thickness was adopted to simulate the contact surface between the anti-seepage wall and the dam [29]; the setting parameters are shown in Tables 2 and 3.

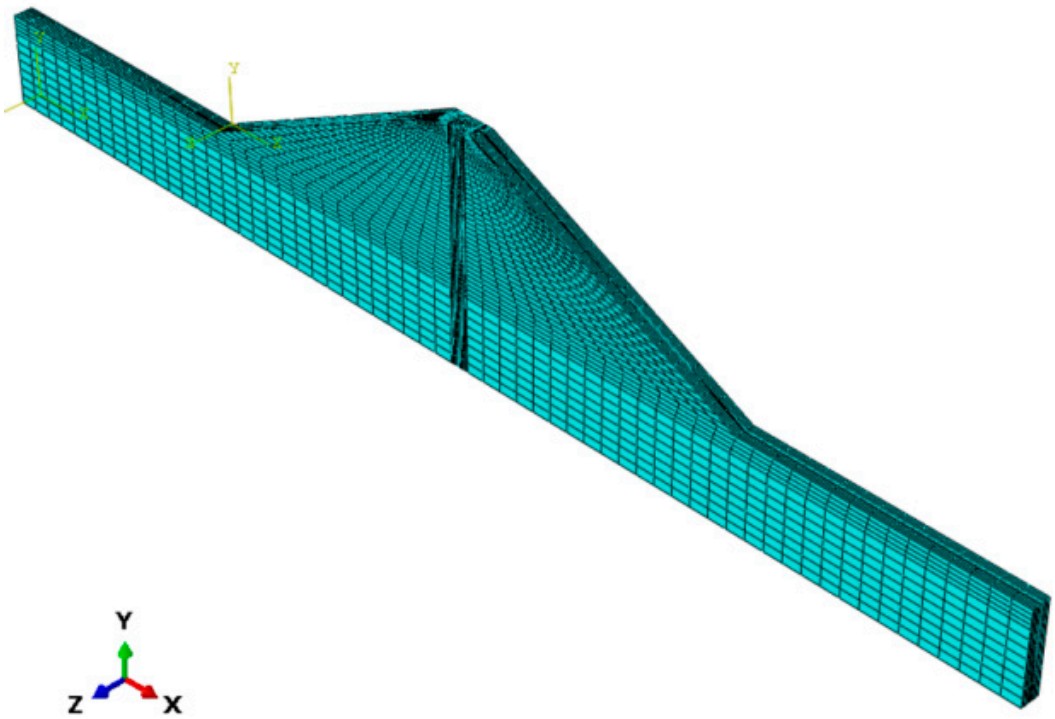

**Figure 1.** Three-dimensional finite element model.

**Table 1.** Material parameters of anti-seepage wall and soil body.

| Material | Density (kg/m³) | Young's Modulus (kPa) | Poisson's Ratio | Permeating Coefficient (cm/s) | Cohesion (kPa) | Internal Friction Angle | Dilatancy Angle |
|---|---|---|---|---|---|---|---|
| Dam | 1930 | 100,000 | 0.29 | $5 \times 10^{-7}$ | 12.38 | 20 | 0 |
| Polymer | 300 | 100,176 | 0.258 | $1 \times 10^{-10}$ | — | — | — |
| Rigid concrete | 2400 | 20,000,000 | 0.3 | $1 \times 8^{-10}$ | — | — | — |
| Plastic concrete | 2400 | 500,000 | 0.25 | $1 \times 8^{-10}$ | — | — | — |

**Table 2.** Parameters of Goodman element between the polymer and the dam body.

| S/N | Parameter | Symbol | Value |
|---|---|---|---|
| 1 | Tangential stiffness coefficient | $K_1$ | 300 |
| 2 | Normal stiffness coefficient | $K_2$ | 300 |
| 3 | Test constant | n | 0.34 |
| 4 | Damage ratio | $R_f$ | 0.95 |
| 5 | Interfacial friction angle | $\delta$ | 11.3 |
| 6 | Water bulk density | $R_w$ | 100 |
| 7 | Atmospheric pressure | $P_a$ | 1000 |

**Table 3.** Parameters of Goodman element between the concrete and the dam body.

| S/N | Parameter | Symbol | Value |
|---|---|---|---|
| 1 | Tangential stiffness coefficient | $K_1$ | 1400 |
| 2 | Normal stiffness coefficient | $K_2$ | 1400 |
| 3 | Test constant | n | 0.35 |
| 4 | Damage ratio | $R_f$ | 0.75 |
| 5 | Interfacial friction angle | $\delta$ | 15 |
| 6 | Water bulk density | $R_w$ | 100 |
| 7 | Atmospheric pressure | $P_a$ | 1000 |

In order to compare the deformation coordination characteristics of the anti-seepage walls with different materials and the dam body, models of polymer, rigid concrete and

plastic concrete walls were established. The thickness of the polymer anti-seepage wall is 0.03 m, and that of the concrete anti-seepage wall is 0.24 m. The total number of model elements is 25,100, and that of nodes is 29,799, in which the number of the units and nodes of the anti-seepage wall are 1400 and 1980, respectively. The intersection of the central axis of the anti-seepage wall and the dam foundation was taken as the origin point for reference. Additionally, it was supposed that the *X*-axis was positive along the boundary line between the dam foundation and the dam body to the far river, and the *Y*-axis was positive along the central line of the anti-seepage wall to the dam crest. The schematic diagram of coordinate system and details of the model dimensions are shown in Figure 2.

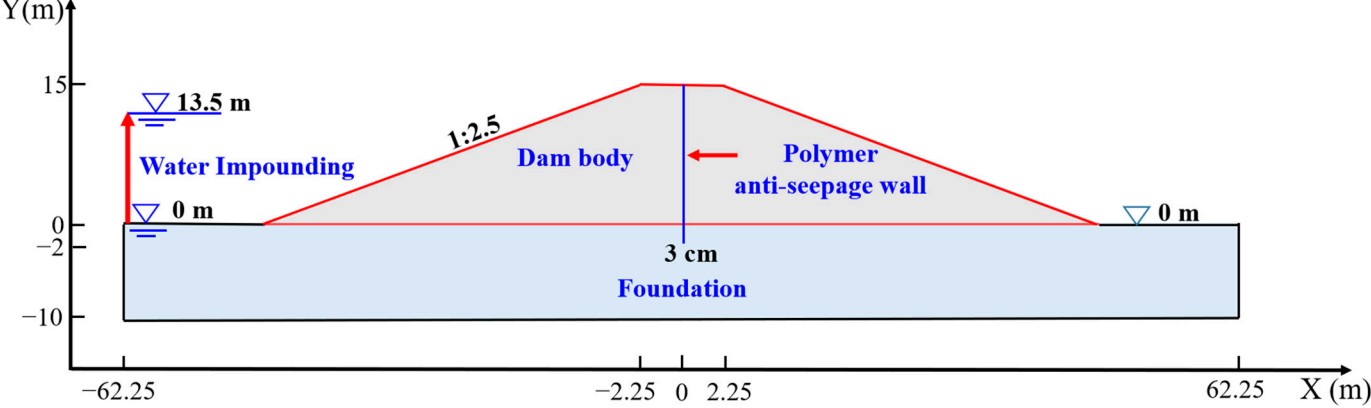

**Figure 2.** Schematic diagram of coordinate system and model dimensions.

### 3.2. Traffic Load

In order to simulate the actual road bearing dynamic load, the traffic load was described by the means of a single half-sine function as shown in Figure 3, and the process of approaching and leaving with standard axle loads was stimulated by adopting the form of impact load. According to The Specification for Design of Highway Asphalt Pavement, trucks with single-axle and double-wheel rear wheels are regarded as standard vehicles, with the standard axle load being 100 kN. Additionally, the traffic load expression is as follows:

$$p(t) = 0.11737 * sin(10\pi t) * 0.7, \ 0 \le t \le 0.1s \tag{3}$$

where *t* refers to the calculation time and $p(t)$ means the traffic load at time *t*. In this paper, the single-circle load was utilized to simulate the rear wheel and double-wheel set of vehicles. The contact surface with ground is shown in Figure 4. The diameter of circle load is 300 mm, the double spacing of wheels is 1.8 m and the grounding area is simplified to two rectangular areas of 0.2 m × 0.3 m at the top of the dam.

During the calculation, ground stress balance was conducted for the calculation model, the water level in front of the dam was set to 13.5 m. The settings of the traffic load amplitude, vehicle speed and driving position are shown in Table 4, and the driving position is represented by the X coordinates of the central line of the axle as shown in Figure 5.

**Table 4.** Condition setting table.

| S/N | Load Amplitude (MPa) | Vehicle Speed (km/h) | Driving Position (m) |
|---|---|---|---|
| 1 | 0.7 | 20 | 1 |
| 2 | 0.9 | 40 | 0.5 |
| 3 | 1.1 | 60 | 0 |
| 4 | 1.3 | 80 | −0.5 |
| 5 | 1.5 | 100 | −1 |

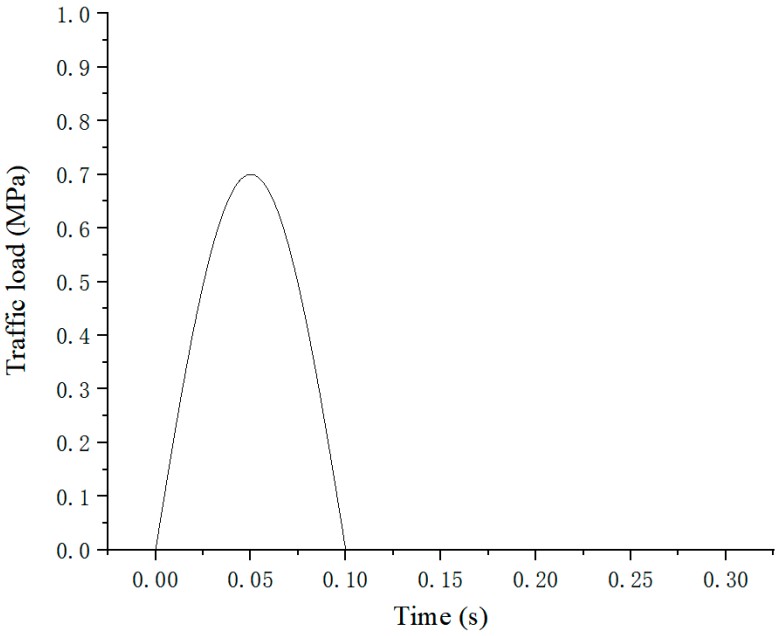

**Figure 3.** Contact surface between tire and ground.

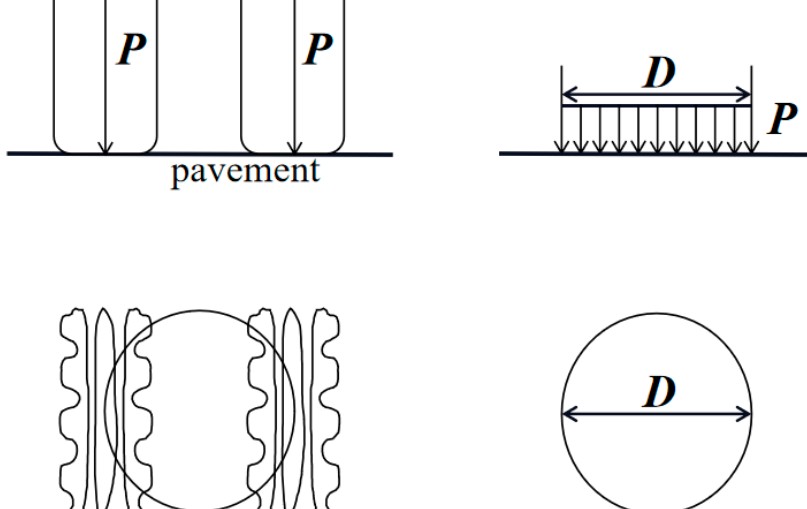

**Figure 4.** Contact surface between tire and ground.

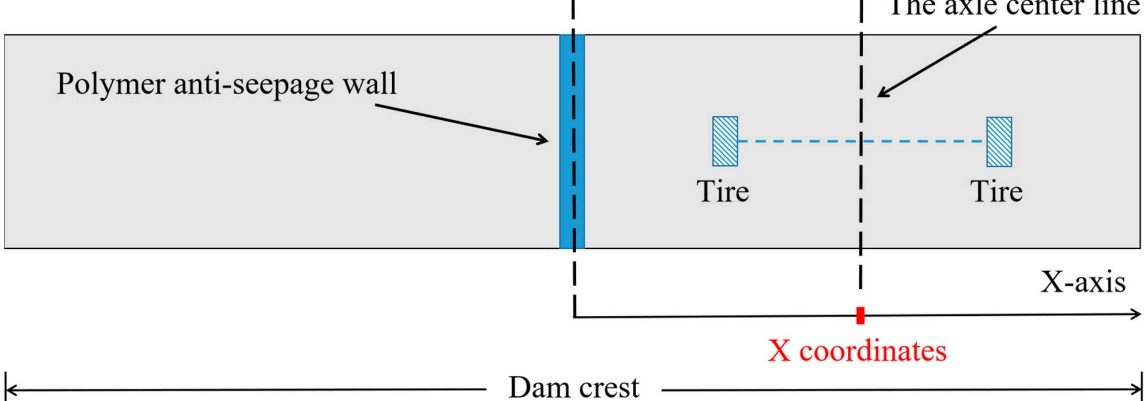

**Figure 5.** Driving position diagram.

### 3.3. Seepage-Stress Coupling Analysis

As a water retaining structure, the stability of the dam is greatly affected by seepage, and the stress deformation of the dam body and seepage is a typical seepage-stress coupling problem. Therefore, the interaction between the seepage wall and the dam body, which is an important part of the dam body, also has seepage-stress coupling problems. This section introduces the principles and methods related to the analysis of seepage stress coupling in dam engineering by combining the seepage-stress coupling function of ABAQUS finite element calculation software.

The dam soil is a porous material, and the pores contain two parts of fluid: liquid and gas. The volume of the soil body includes both soil particles and pores. Since the tensile stress is positive and the compressive stress is negative in the ABAQUS calculation, but both gas pressure and liquid pressure are positive in terms of pressure, the effective stress principle expression is:

$$\bar{s} = s + (\mu u_w + (1 - \mu)\mu_a) \tag{4}$$

where $\bar{s}$ is the effective stress, $s$ is the total stress, $\mu$ is the effective stress parameter, $u_w$ is the liquid pressure and $u_a$ is the gas pressure, when the soil is fully saturated, $\mu = 1$ and when the soil is completely dry, $\mu = 0$.

The effective stress of the material in ABAQUS is the basis of the stress–strain relationship and calculation results. The pore water pressure in the coupled seepage-stress analysis can be chosen based on the total pore pressure or super pore pressure, and in this paper, the gravity applied load is used, which means the analysis is based on the total pore pressure.

The permeability law used in ABAQUS is the Forchheimer equation, and the permeability coefficient is expressed as:

$$\bar{k} = \frac{k_s}{(1 + \beta\sqrt{v_w v_w})}k \tag{5}$$

where $k$ is the seepage coefficient of saturated soil; $v_w$ is the flow speed, $\beta$ is the parameter reflecting the effect of flow speed on the seepage coefficient, when $\beta = 0$; Equation (5) is simplified to Darcy formula; $k_s$ is the saturation coefficient, when the soil is saturated, $k_s = 1$, and the coefficient $k_s$ reflects the relationship between the permeability coefficient of non-saturated soil and saturated soil.

In addition, the permeability coefficient $k$ can also be a function related to the pore ratio, characterizing the effect of soil skeleton changes on the permeability coefficient, and the effect of the stress field on the seepage field can be calculated by setting the relationship function between the permeability coefficient and the pore ratio.

According to Equation (5), the effect of saturation on permeability coefficient can be reflected by $k_s$. In ABAQUS, when saturation $S_r < 1$, $k_s = S_r^3$, and $k_s = 1$ when $S_r \geq 1$. For the coupling effect of seepage and stress in unsaturated soil, it is realized by the relationship between negative pore pressure and permeability coefficient, and in saturated soil, it is realized by setting the function between permeability coefficient and pore ratio. Due to the hygroscopic and dewatering characteristics of unsaturated soils, it is necessary to perform unsaturated soil seepage and stress calculations to determine the relationship between hygroscopicity and dewatering and the permeability coefficient

Based on the above analysis, it can be seen that the matrix suction in unsaturated soil is related to the permeability coefficient, realizing the interaction between seepage flow and stress in unsaturated soil.

The coupled seepage seepage-stress analysis needs to unify the saturated and unsaturated seepage calculations. To simplify the calculation, it is assumed in this paper that both saturated and unsaturated soils obey the law, ignore the effect of flow velocity, and distinguish by permeability coefficient and pore water pressure, and the C3D8P element with pore pressure degree of freedom was used for the solution. The water pressure in the pores is shown in Figure 6.

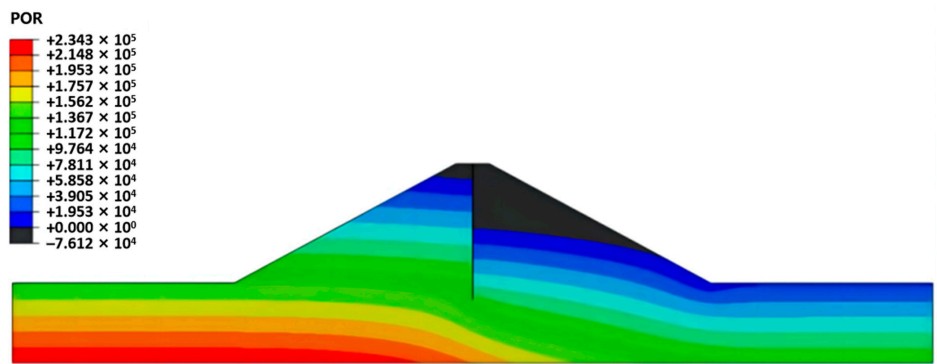

**Figure 6.** The water pressure in the pores.

## 4. Calculation Results

*4.1. Comparison of Deformation Coordination Characteristics between the Anti-Seepage Wall with Different Materials and the Dam Body*

In order to study the deformation coordination characteristics between the wall with different materials and the dam, the vehicle speed of 20 km/h was taken as an example. Specifically, when vehicles drive above the anti-seepage wall, traffic load with an amplitude of 0.7 MPa was applied, and the anti-seepage walls with different materials were adopted. As shown in Figure 7, the variation trend of the displacement difference between the concrete anti-seepage wall and the dam body is basically the same along the wall height, that is, it gradually decreases from dam crest downward. While, the displacement difference between the dam body and the rigid concrete anti-seepage wall at each height is much larger than that between the dam body and the plastic concrete anti-seepage wall, the maximum values of the displacement difference appear at the dam crest, which are 1.6 cm and 0.58 cm, respectively; the displacement difference between the polymer anti-seepage wall and the dam body gradually increases along the dam crest downward, reaching a maximum value of 0.18 cm at a wall height of 5 m, and gradually decreases. From the perspective of the maximum displacement difference, the rigid concrete anti-seepage wall is larger, and the polymer anti-seepage wall is smaller. Hence, it can be seen that the deformation morphological characteristics of polymer anti-seepage wall and the dam body are better than that of the rigid and plastic concrete anti-seepage walls.

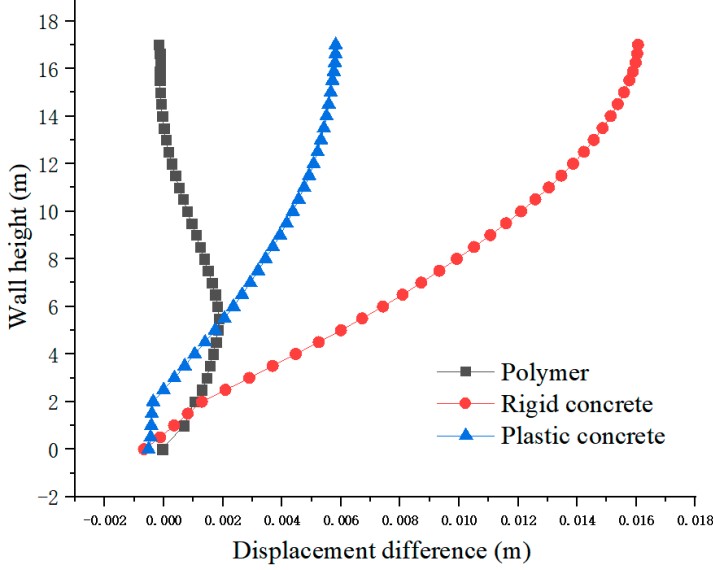

**Figure 7.** Comparison of displacement difference of anti-seepage walls with different materials along the wall.

### 4.2. Deformation Coordination Characteristics between Polymer Anti-Seepage Wall and Dam Body

#### 4.2.1. Influence of Amplitude

In order to determine the influence depth of traffic load, the vehicle speed of 20 km/h was taken as an example. In detail, the subsidence, Mises stress and displacement difference distribution of the dam seepage wall along the wall height under different load amplitude conditions when vehicles drive above the polymer wall are shown in Figures 8–10.

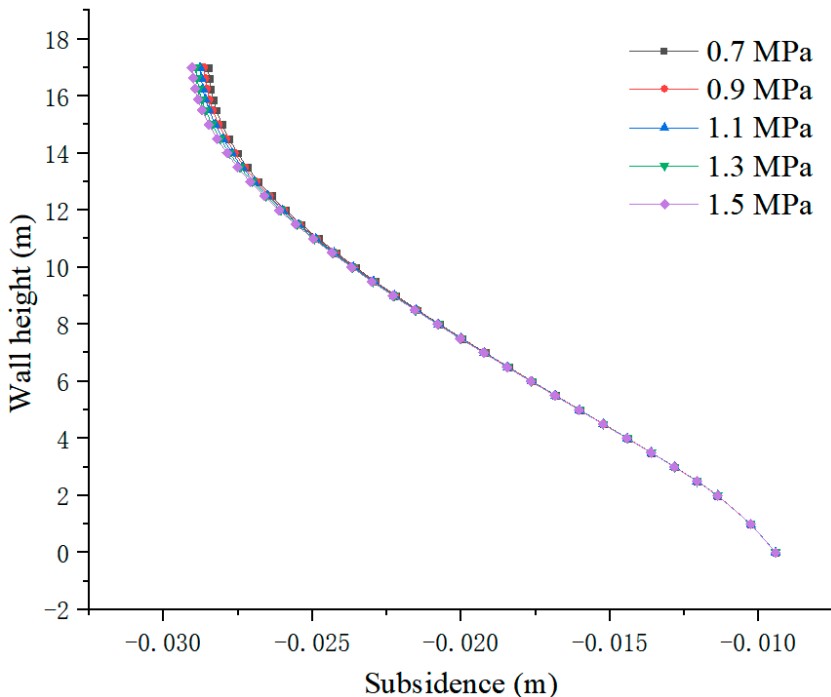

**Figure 8.** Distribution of subsidence along the anti-seepage wall under the action of different load amplitudes.

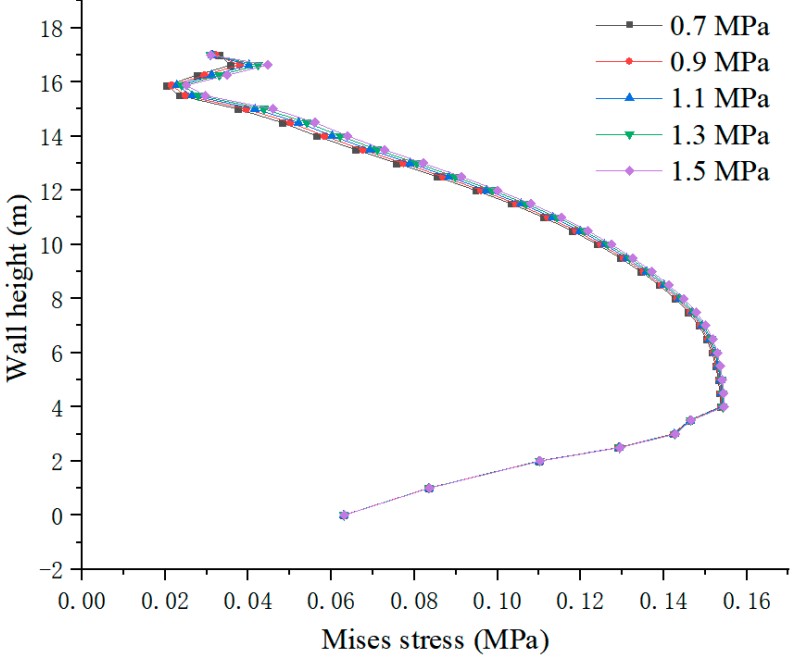

**Figure 9.** Distribution of Mises stress along the anti-seepage wall under different load amplitudes.

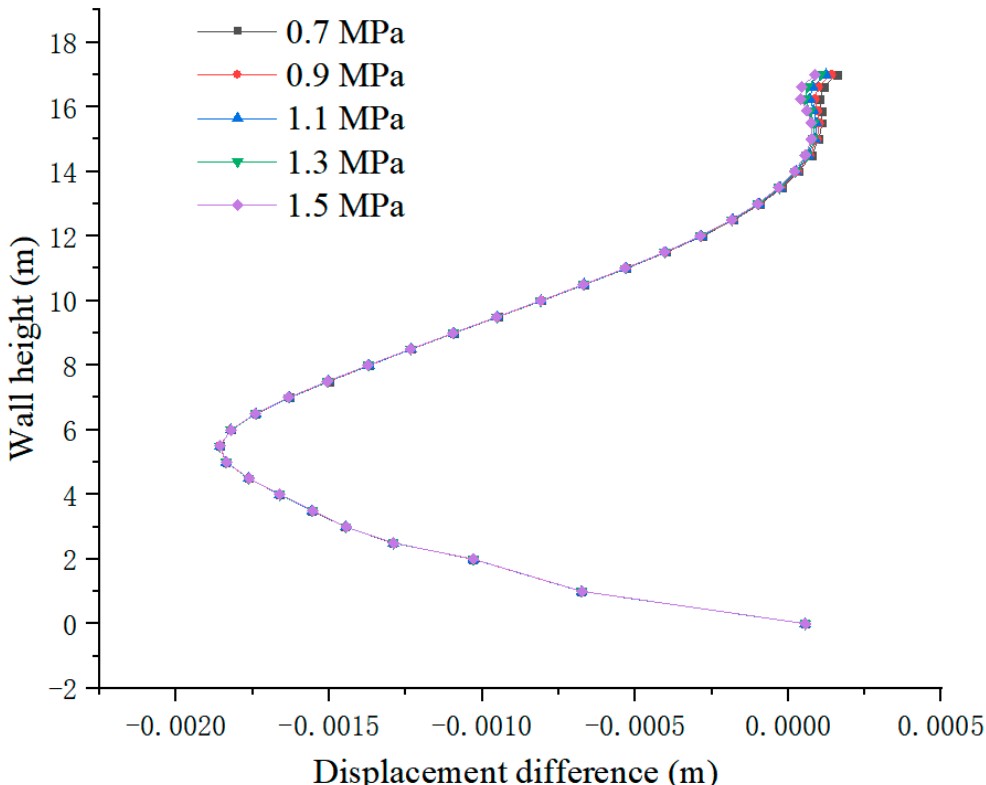

**Figure 10.** Distribution of the displacement difference along the anti-seepage wall under different load amplitudes.

As shown in Figures 8 and 10, the maximum position of the change amplitude of the subsidence and the displacement difference are at the wall height of 16.625 m. Figure 9 shows that in the range of 0–4 m, the Mises stress gradually increases with the height of the wall. Additionally, in the range of 4–15 m, it gradually decreases with the increase of the wall height. In the range of 15–17 m, a certain fluctuation occurs, the greatest variation appearing at the wall height of 16.625 m. To sum up, under the traffic load and pore pressure, the most responsive part of the dam is in the middle dam, but the part that is more sensitive to traffic load is at the dam crest. The influence depth of traffic load on the dam strain is about 5 m. The height of 16.625 m should be chosen, that is, the dam height of 14.625 m is the research section.

In order to analyze the variation law of dam subsidence with the amplitude of traffic load, when vehicles drive above the polymer wall, the vehicle speed of 20 km/h was taken as an example—Figure 11 shows the subsidence of the dam body and the anti-seepage wall under different load amplitudes. It can be seen that under traffic load, the subsidence curve presents "saddle-shaped double peak", an obvious drop in subsidence at the junction occurs, and the maximum subsidence of the dam body reaches about 2.98 cm. Under the same conditions, with the increase of the load amplitude, the subsidence of the dam body and the wall gradually increases. The subsidence difference of the dam body at the left wheel under the load amplitudes from 0.7 to 1.5 MPa, reaching 1.1 mm, with the amplitude being 3.7%. The cloud diagrams of wall subsidence and Mises stress under different load amplitudes are as follows.

The subsidence curves of the wall at the dam height of 14.625 m under different load amplitudes when the vehicle travels above the polymer wall at a speed of 20 km/h are shown in Figure 12. It can be seen that, under the same conditions, the subsidence of the wall increases with the increase of amplitude, and the subsidence difference of wall under the load amplitudes from 0.7 to 1.5 MPa is 0.55 mm, with an amplitude of variation of 1.7%.

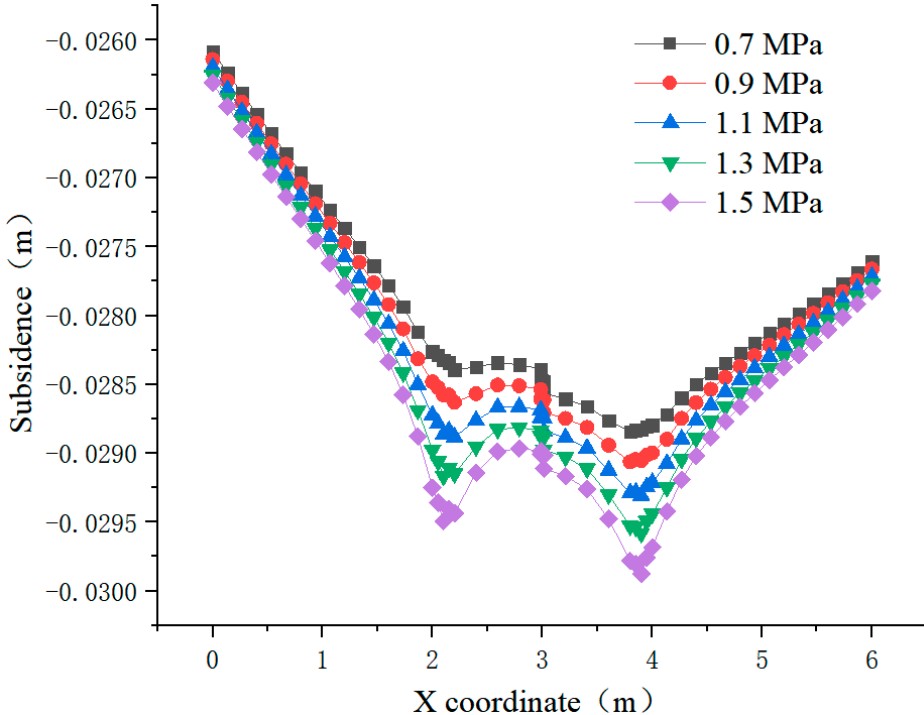

**Figure 11.** Distribution of dam subsidence at 14.625 m under different load amplitudes.

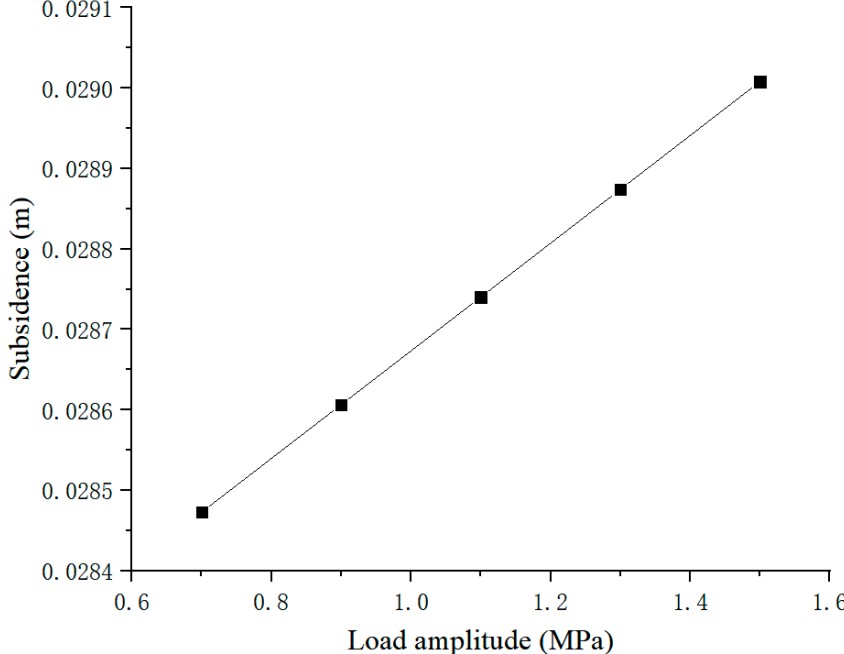

**Figure 12.** Variation of wall subsidence at 14.625 m under different load amplitudes.

In order to analyze the variation law of Mises stress of the dam with the amplitude of the traffic load, the vehicle speed of 20 km/h was taken as an example. In detail, the Mises stress of the dam body and the anti-seepage wall under different load amplitude conditions when vehicles drive above the polymer wall are shown in Figure 13. It can be seen that the Mises stress of the wall has an obvious peak value at the wheel, and the maximum Mises stress reaches 0.23 MPa. Under the same conditions, with the increase of load amplitude, the Mises stress of the dam and the wall gradually increases, with considerable change. Additionally, the difference of the Mises stress of dam at the wheel point under the load

amplitudes from 0.7 to 1.5 MPa reaches 0.063 MPa, with the amplitude of variation of 27.8%, indicating that the traffic load amplitude has a significant impact on Mises stress of the dam.

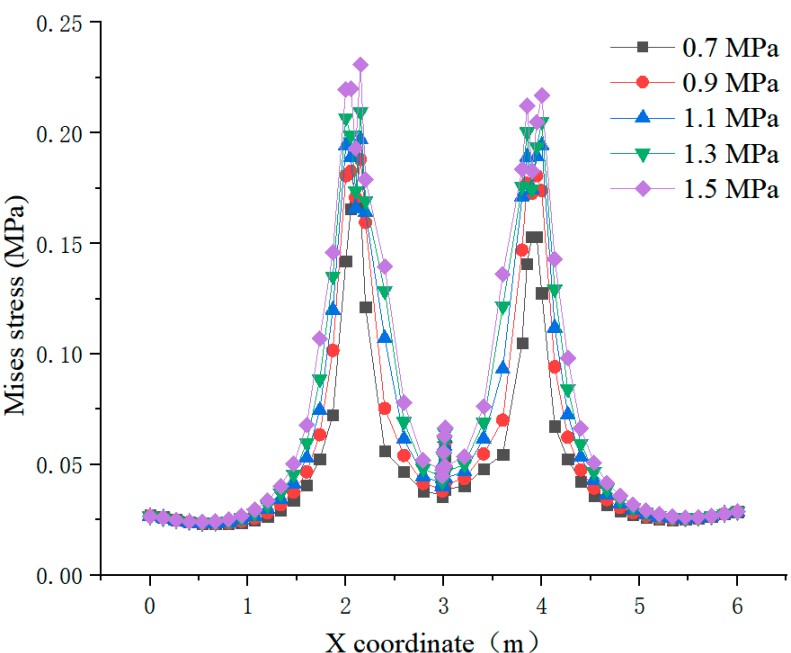

**Figure 13.** Distribution of dam Mises stress at 14.625 m under different load amplitudes.

The Mises stress curves of the wall at the dam height of 14.625 m under different load amplitudes when the vehicle travels above the polymer wall at a speed of 20 km/h are shown in Figure 14. It can be seen that, under the same conditions, the Mises stress of the wall grows with the increase of amplitude. The difference of the Mises stress of the wall under the load amplitudes from 0.7 to 1.5 MPa is 8996 Pa, with the amplitude of variation of 25%, indicating that the traffic load amplitude exerts a significant effect on the Mises stress of the wall.

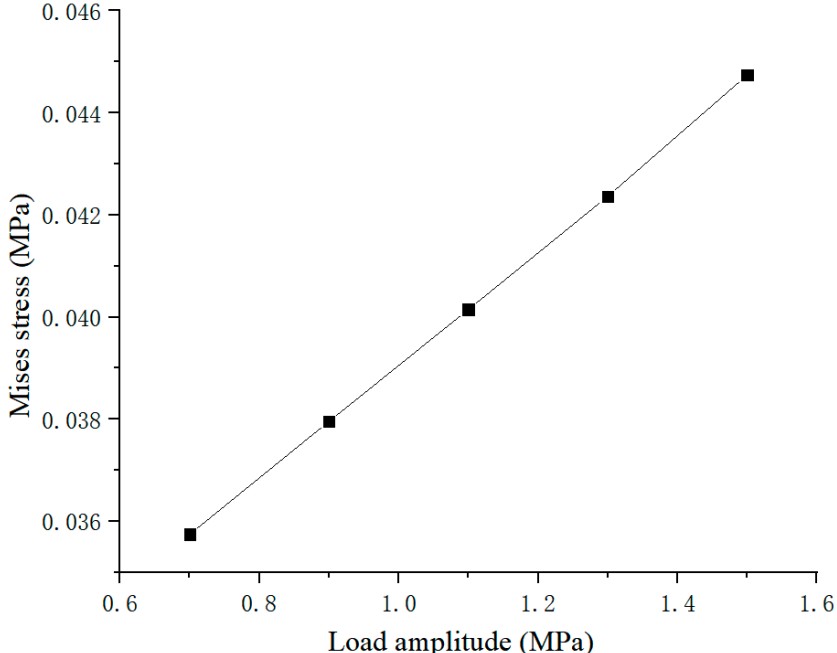

**Figure 14.** Variation of wall Mises stress at 14.625 m under different load amplitudes.

In order to analyze the variation law of dam displacement difference with traffic load amplitude, vehicle speed of 20 km/h was taken as an example. The displacement difference of the dam and the anti-seepage wall under different load amplitudes when vehicles drive above the polymer wall are shown in Figure 15. It can be seen that the displacement difference on both sides of the dam is symmetrical on the central line of the dam, as there is a wheel load equal to the wall on both sides of dam. Additionally, the displacement difference of the wall shows obvious peak value under the four edges of wheel action, with the maximum displacement difference of 0.37 mm. Under the same conditions, with the rise of the load amplitude, the displacement difference between the dam and the wall also gradually increases, with more obvious change among them. The displacement difference of the dam at the edge of the left wheel under the load amplitudes from 0.7 to 1.5 MPa is 0.23 mm, with the amplitude of variation of 62.2%, indicating that the amplitude has a significant impact on the displacement difference of the dam.

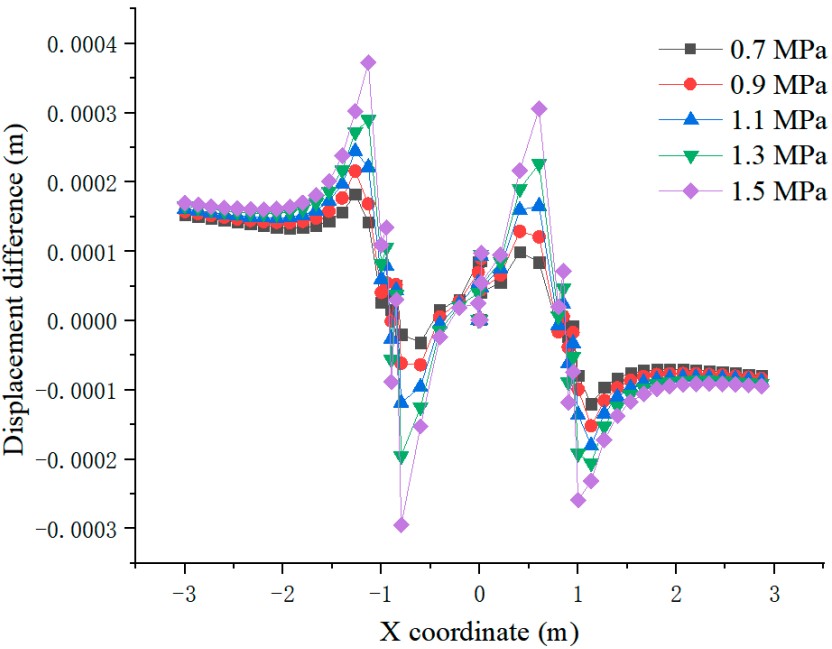

**Figure 15.** Distribution of displacement difference at 14.625 m under different load amplitudes.

When the vehicle travels above the polymer wall at a speed of 20 km/h, the displacement difference curve of the wall at the dam height of 14.625 m under different load amplitudes is shown in Figure 16. It can be observed that under the same conditions, the displacement difference enhances with the increase of the amplitude, and the displacement difference under the load amplitudes from 0.7 to 1.5 MPa is 0.061 mm, with the amplitude of variation of 87.1%.

It shows that the amplitude of traffic load exerts a vital impact on the deformation coordination between the polymer anti-seepage wall and the dam. The vertical subsidence, Mises stress and displacement difference of the dam rises with the increase of traffic load amplitude. Therefore, in terms of the dam with traffic load, the lower the traffic load amplitude is, the more conducive it is to the coordinated deformation between the dam and the polymer anti-seepage wall. In order to protect the dam, the vehicles crossing the dam should be strictly limited in weight, and overweight vehicles should be prohibited from boarding the dam.

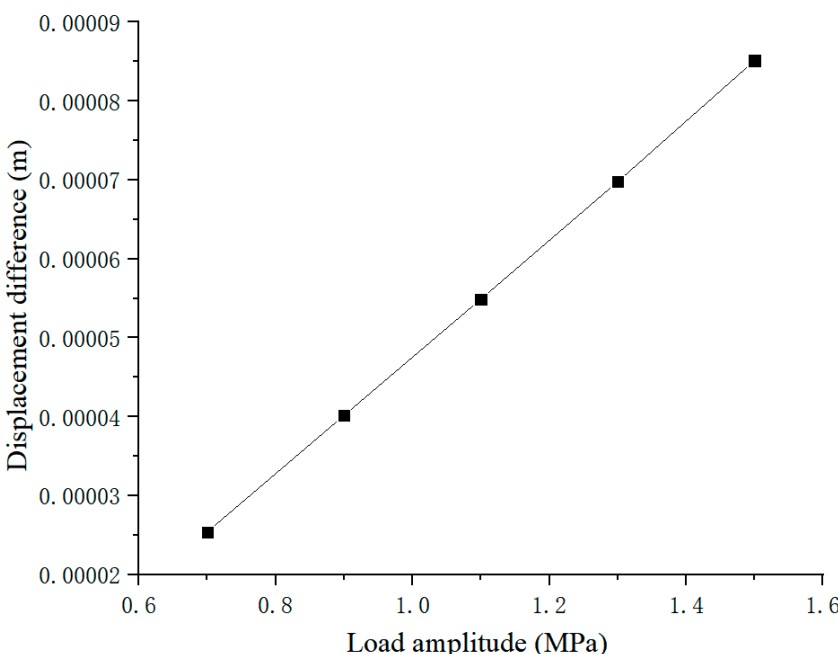

**Figure 16.** Variation of displacement difference at 14.625 m under different load amplitudes.

### 4.2.2. Influence of Vehicle Speed

In order to analyze the variation law of dam subsidence stress with the vehicle speed, when vehicles drive above the polymer wall, the vehicle load amplitude of 0.7 MPa was taken as an example. Figure 17 shows the response of the dam body and the anti-seepage wall under different vehicle speeds. It can be observed that under traffic load, the subsidence curve presents "saddle-shaped double peak", and an obvious drop in subsidence at the junction occurs, with the maximum subsidence of the dam body reaching about 2.85 cm. Under the same conditions, with the acceleration of the vehicle speed, the subsidence of the dam and the wall gradually decreases. The subsidence difference of the dam at the left wheel under the speed from 20 to 100 km/h is 0.5 mm, with the amplitude of variation of 1.8%.

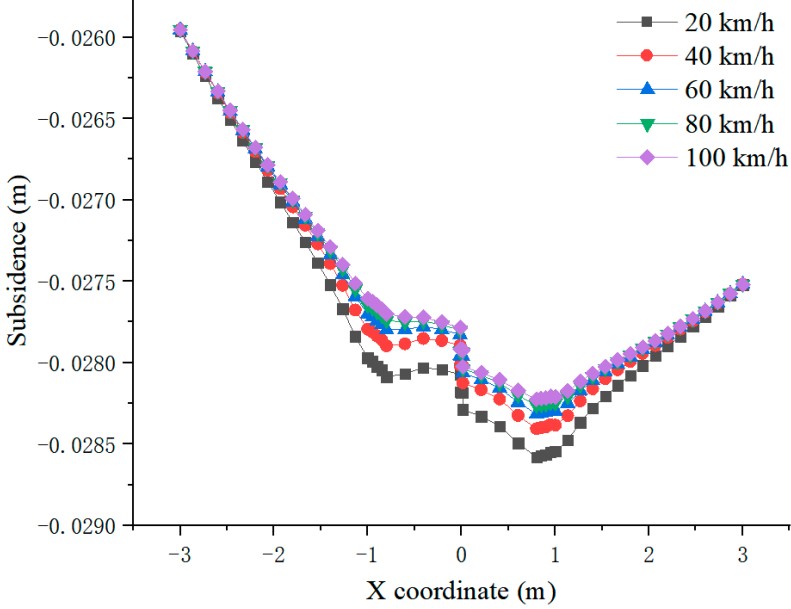

**Figure 17.** Distribution of dam subsidence at 14.625 m under different vehicle speeds.

The subsidence curve of the wall at the dam height of 14.625 m under different vehicle speeds when the vehicle with load amplitude of 0.7 MPa drives above the polymer wall is shown in Figure 18. It shows that under the same conditions, the subsidence of the wall decreases with the increase of the vehicle speed. The subsidence difference of the wall under the speed from 20 to 100 km/h is 0.27 mm, with the amplitude of variation of 0.89%.

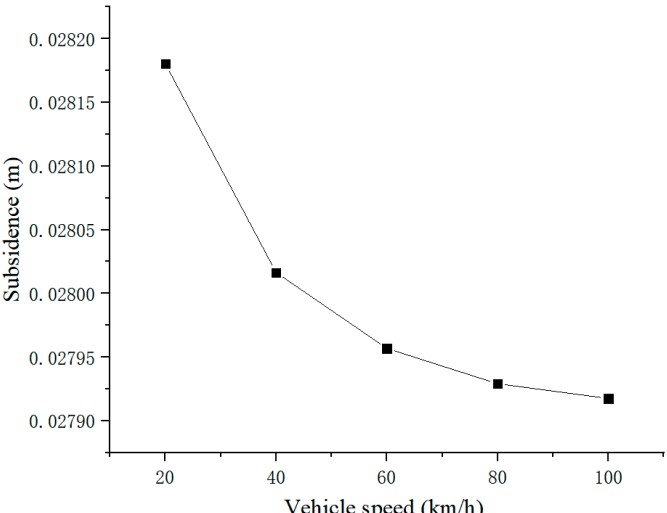

**Figure 18.** Variation of wall subsidence at 14.625 m under different vehicle speeds.

In order to analyze the variation law of Mises stress of the dam with the vehicle speed, the amplitude of vehicle load of 0.7 MPa/h was taken as an example. The Mises stress of the dam body and the anti-seepage wall under different vehicle speeds when vehicles drove above the polymer wall is shown in Figure 19. It can be seen that the subsidence of the wall has an obvious peak value at the wheel, with the maximum Mises stress reaching 0.16 MPa. Under the same conditions, with the acceleration of the vehicle speed, the Mises stress of the dam and the wall gradually decreases, with considerable change. The difference of Mises stress of the dam body at the left wheel under the vehicle speed from 20 to 100 km/h is 0.072 MPa, with the amplitude of variation of 45.9%.

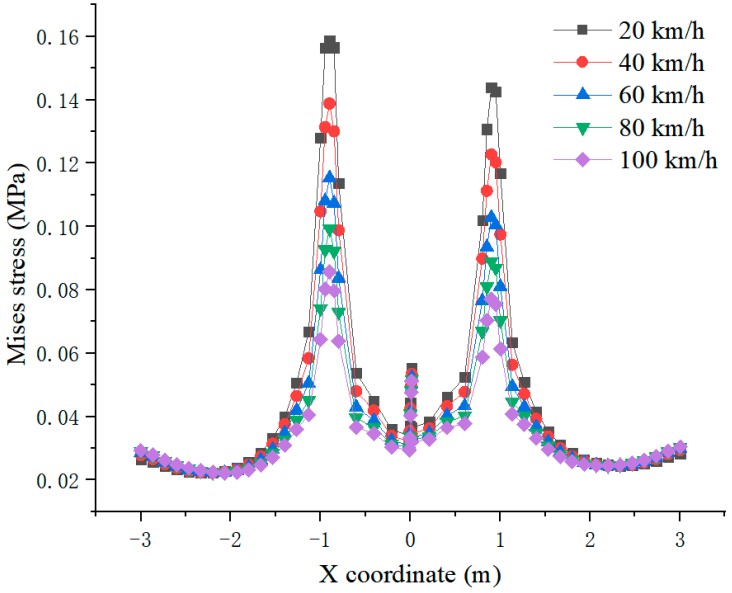

**Figure 19.** Distribution of Mises stress of the dam at 14.625 m under different vehicle speeds.

The Mises stress curve of the wall at the dam height of 14.625 m under different vehicle speeds when the vehicles drove above the polymer wall with the load amplitude of 0.7 MPa is shown in Figure 20. It shows that under the same conditions, the Mises stress of the wall decreases with the acceleration of the vehicle speed. The difference of the Mises stress of the wall under the vehicle speed from 20 to 100 km/h is 4001 Pa, with the amplitude of variation of 11.9%.

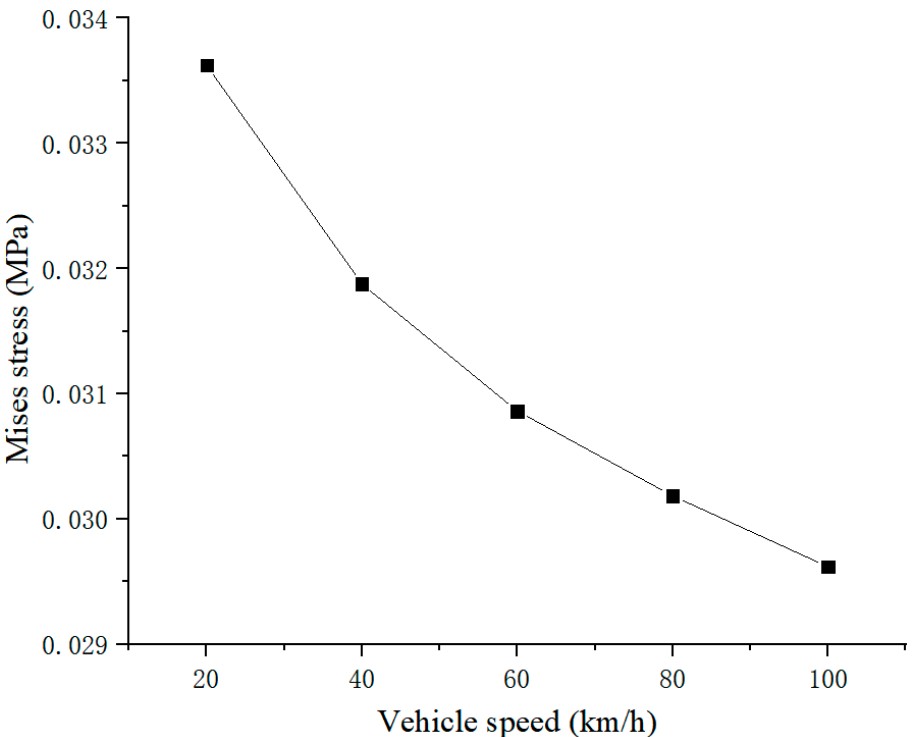

**Figure 20.** Variation of Mises stress of the wall at 14.625 m under different vehicle speeds.

The displacement difference of the dam and the anti-seepage wall under different vehicle speeds when vehicles drive above the polymer wall with the vehicle load amplitude of 0.7 MPa is shown in Figure 21. It can be observed that the displacement difference on both sides of the dam is symmetrical on the central line of the dam, as there is a wheel load equal to the wall on both sides of the dam. The displacement difference of the wall shows obvious peak value under the four edges of wheel action, with the maximum displacement difference of 0.37 mm. Under the same conditions, with the acceleration of the vehicle speed, the displacement difference of the dam gradually decreases. The displacement difference of the dam at the left wheel edge under the speed from 20 to 100 km/h is 0.04 mm, with the amplitude of variation of 38.5%.

The displacement difference curve of the wall at the dam height of 14.625 m under different load amplitudes when the vehicles drove above the polymer wall with the load amplitude of 0.7 MPa is shown in Figure 22. It can be discovered that under the same conditions, the subsidence of the wall rises with the increase of the vehicle speed. The subsidence difference of the wall under the speed from 20 to 100 km/h is 0.19 mm, with the amplitude of variation of 17.8%.

It can be seen from the above analysis that under the same conditions, the deformation coordination between the polymer anti-seepage wall and the dam is affected greatly by different vehicle speeds. The faster the vehicles drive, the smaller the vertical subsidence and Mises stress of the wall are. Additionally, the displacement difference between the wall and the dam grows with the increase of the vehicle speed. In consideration of the small value of the displacement difference, for the dam bearing traffic load, without considering

the speed limit, the faster the vehicles drive, the more conducive it is to the coordination between the dam and the polymer anti-seepage wall.

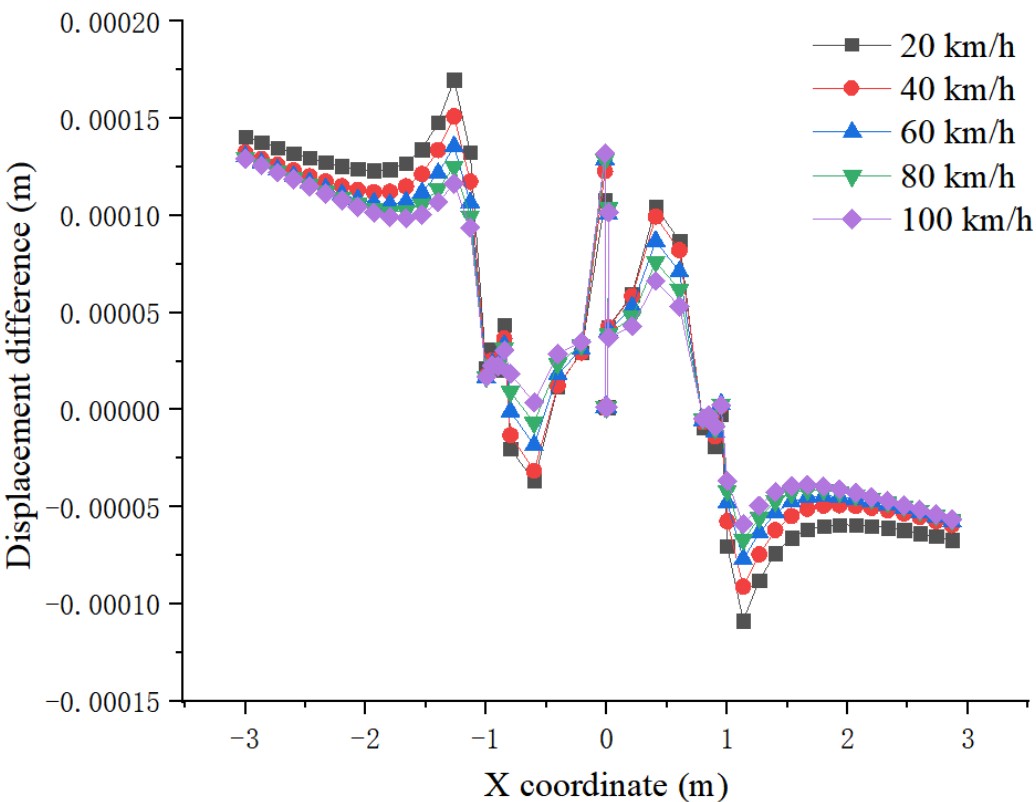

**Figure 21.** Distribution of displacement difference at 14.625 m under different vehicle speeds.

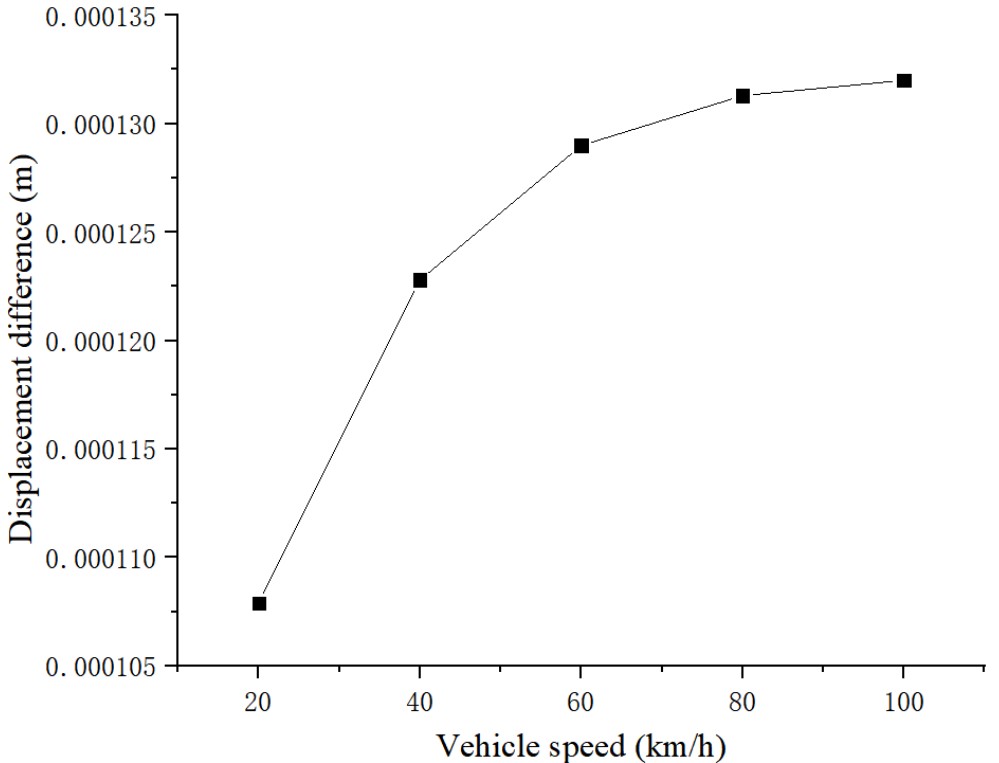

**Figure 22.** Variation of displacement difference at 14.625 m under different vehicle speeds.

### 4.2.3. Influence of Driving Position

In order to analyze the variation law of dam subsidence with the driving position, the vehicle speed of 20 km/h was taken as an example. The subsidence of the dam and the anti-seepage wall at different driving positions when the amplitude reaches 0.7 MPa is shown in Figure 23. It can be seen that there is a significant difference in the subsidence at the junction of the dam and the wall. The influence trend of the driving position on the subsidence of the dam has a certain symmetry on both sides of the dam. Vehicles on the left side of the dam exert more significant effects than these on the right side with the same distance. This results from the higher pore water pressure on the left side and the coupling of seepage stress. When the vehicle acts on both sides of the dam, the maximum value of the subsidence of the dam appears at the junction of the dam and the anti-seepage wall. When the vehicle acts on the middle of the dam, the peak value of subsidence appears at the wheel, and the peak value at this time is the minimum value of 2.86 cm among the peaks of each working condition. The maximum peak value appears in the 0.5 m wheelbase group, with the maximum value of 2.91 cm, and the variation range of 1.7%

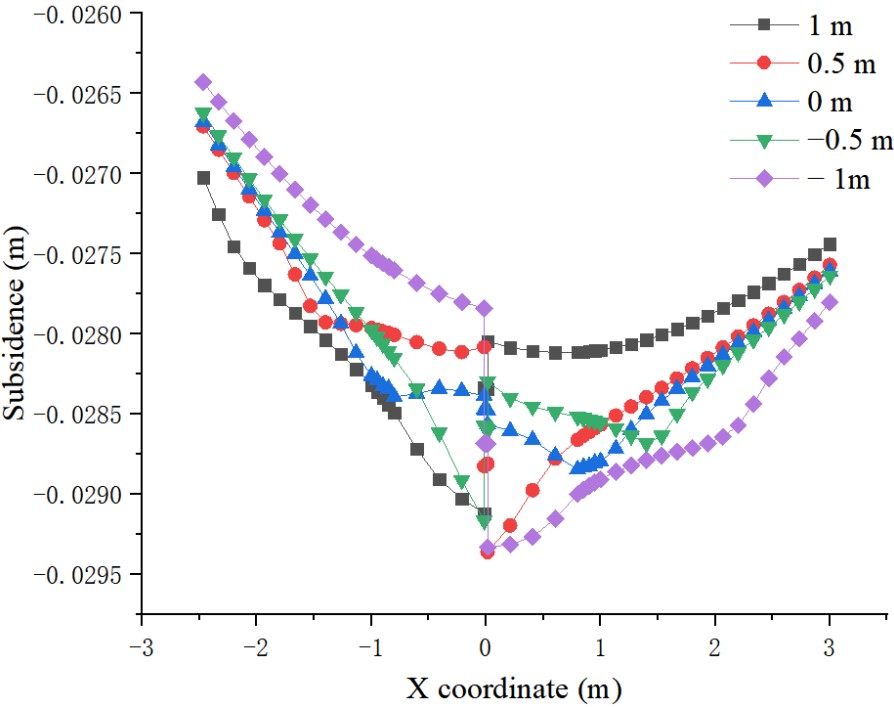

**Figure 23.** Distribution of dam subsidence at 14.625 m under different load positions.

The subsidence curve of the wall at dam height of 14.625 m at different driving positions when the vehicles drove with the load amplitude of 0.7 MPa and the vehicle speed of 20 km/s is shown in Figure 24. It can be seen that under the same conditions, from the far right to the far left of the dam, the influence of driving load gradually decreases, and a peak value of 2.88 cm appears at the wheelbase of 0.5 m, with an amplitude of variation of 1.6%. For the subsidence of wall, the most unfavorable driving position is near the wheelbase of 0.5 m in the middle of the dam.

In order to analyze the variation law of the Mises stress of the dam with the driving position, the vehicle speed of 20 km/h was taken as an example. The Mises stress of the dam and the anti-seepage wall at different driving positions when the amplitude is 0.7 MPa, is shown in Figure 25. It can be seen that the influence trend of the driving position on the Mises stress of the dam has a certain symmetry on both sides of the dam. The maximum peak value is 0.19 MPa at the wheel on the right side. Additionally, when the vehicle acts on the dam, the peak value of the Mises stress is the smallest, with the minimum peak value of 0.062 MPa.

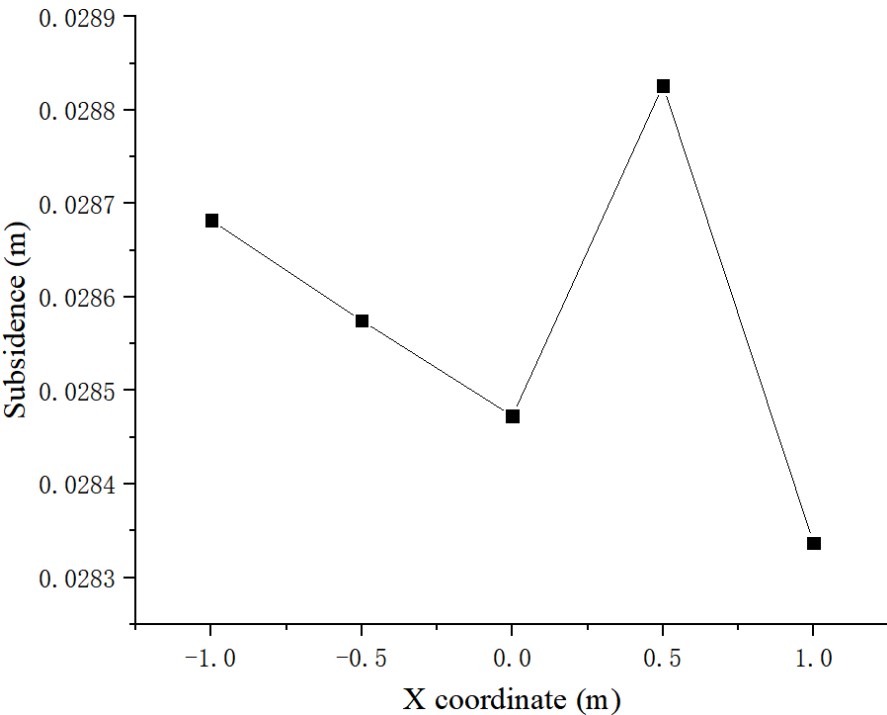

**Figure 24.** Variation of wall subsidence at 14.625 m under different load positions.

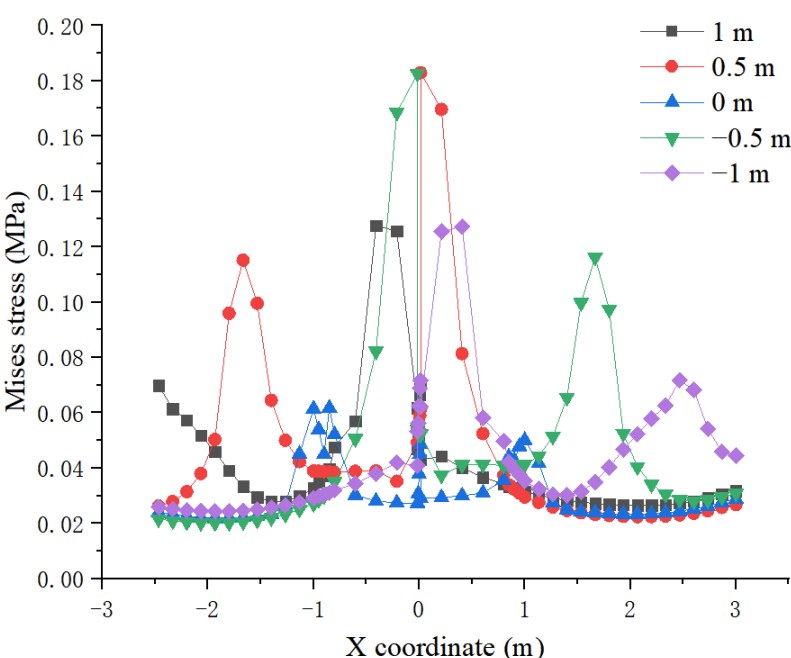

**Figure 25.** Mises stress distribution of the dam at the height of 14.625 m under different driving positions.

The Mises stress curve of the wall at the dam height of 14.625 m at different driving positions when the vehicles drove with the load amplitude of 0.7 MPa and the vehicle speed of 20 km/s is shown in Figure 26. It can be seen that under the same conditions, from the far right to the far left of the dam, the influence of traffic load remains within a certain range. However, the influence is significantly reduced when the load is on the center of the dam. Meanwhile, the Mises stress is 0.27 MPa, and the maximum value appears at the wheelbase of 0.5 m, the Mises stress at this time is 0.54 MPa, with the variation range of 50%. Therefore, with regard to the Mises stress of the wall, the best driving position is near

the middle of the dam crest, and the most unfavorable driving position is near the 0.5 m near the water side of the dam.

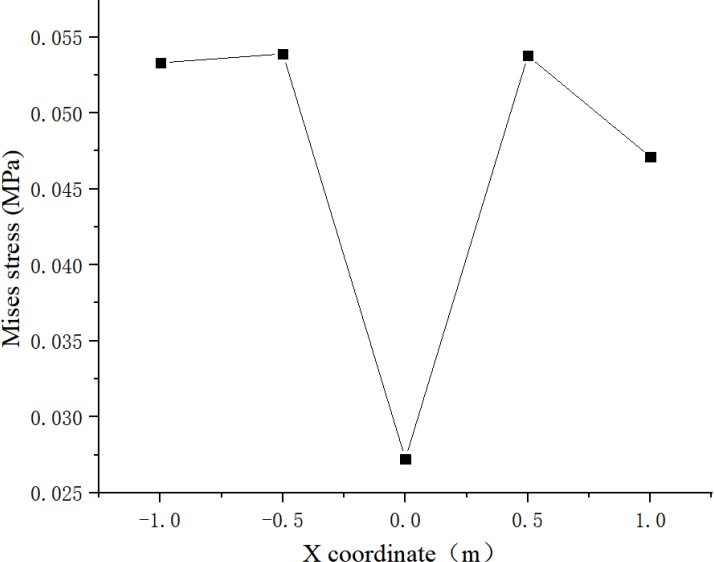

**Figure 26.** Variation of wall Mises stress at 14.625 m under different load positions.

In order to analyze the variation law of displacement difference stress of the dam with the driving position, the vehicle speed of 20 km/h was taken as an example. The displacement difference of the dam and the anti-seepage wall at different driving positions when the amplitude is 0.7 MPa is shown in Figure 27. It can be seen that the influence trend of the driving position on the subsidence of the dam has a certain symmetry on both sides of the dam, with significant fluctuations. When the vehicle acts on the middle of the dam crest, the peak value of the displacement difference is 0.085 mm, which is the minimum value among the peak values of various working conditions. The maximum peak appears in the −1 m wheelbase group, namely, 0.84 mm, with the variation range of 89.9%.

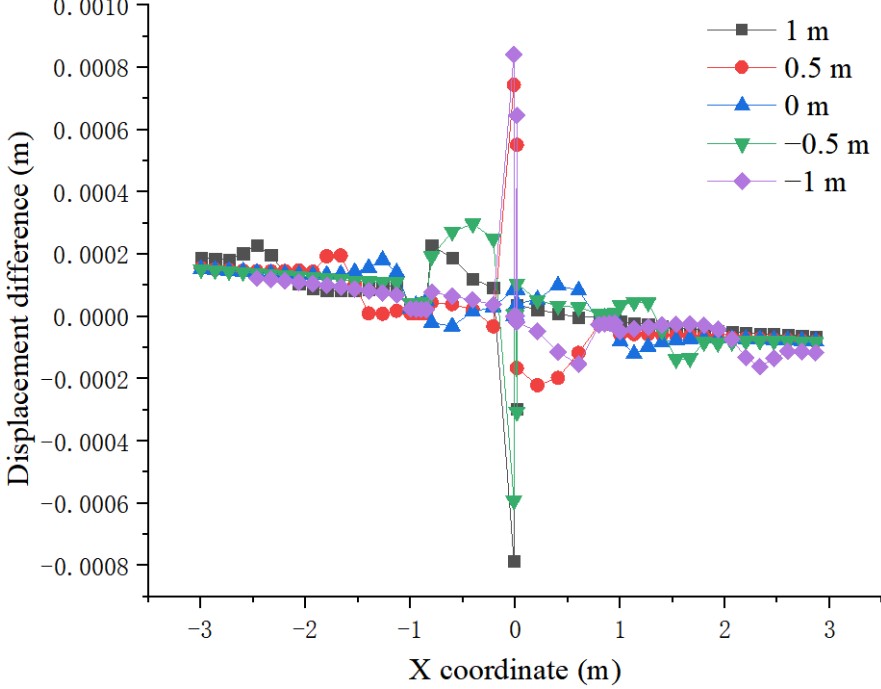

**Figure 27.** Distribution of displacement difference at 14.625 m under different load positions.

The displacement difference curve of the wall at the dam height of 14.625 m at different driving positions when the vehicles drove with the load amplitude of 0.7 MPa and the vehicle speed of 20 km/s is shown in Figure 28. It can be seen that under the same conditions, from the far right to the far left of the dam, the influence of the driving load on the displacement difference between the wall and the dam remains in a certain range, which is significantly reduced when the load acts on the dam, and the displacement difference is 0.085 mm at this time. The maximum value appears when the wheelbase is −1 m, with the displacement difference of 0.84 mm, and change range between the maximum value and the minimum value is 90.3%. Thus, it can be seen that for the Mises stress of the wall, the driving position with less influence is in the middle of the dam crest, and that with greater influence is near 1 m on the far water side of the dam.

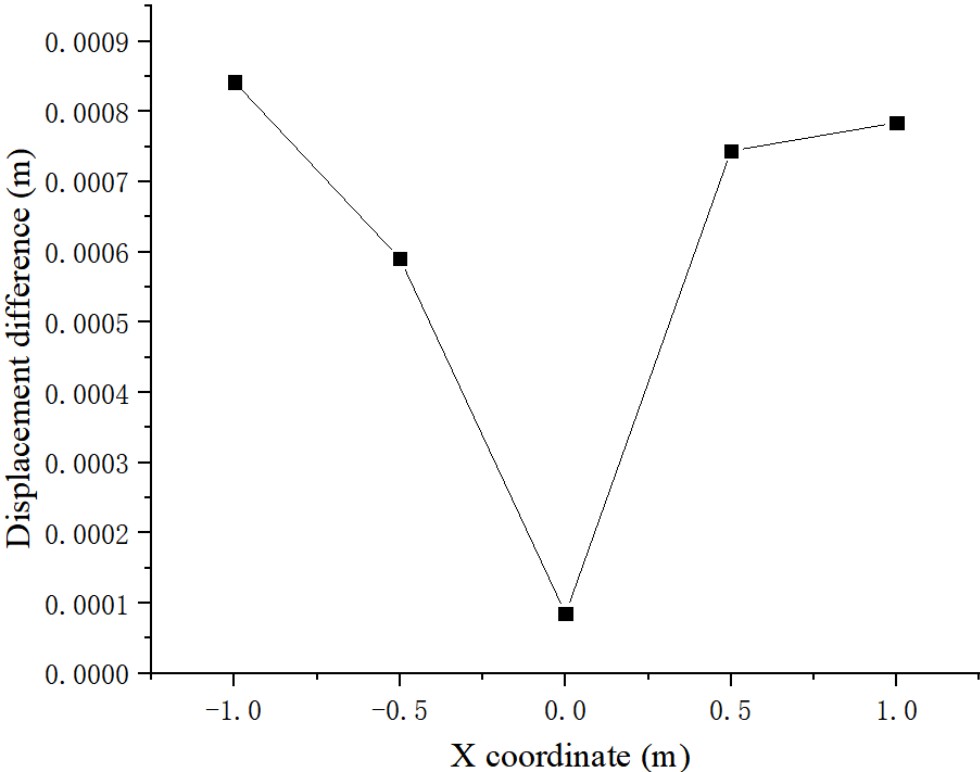

**Figure 28.** Variation of displacement difference at 14.625 m under different load positions.

According to the above analysis, it is known that the driving position has a significant influence on the deformation coordination characteristics between the polymer anti-seepage wall and the dam. When vehicles drive above the anti-seepage wall, the subsidence of the dam, the Mises stress and the deformation difference are all at the lowest or lower level. Regarding the dam body bearing traffic load, when vehicles drive above the anti-seepage wall, it is more conducive to the coordinated deformation between the dam and the polymer anti-seepage wall. Therefore, when designing the driveway at the top of the dam, it should be designed directly above the anti-seepage wall.

## 5. Conclusions

A 3D finite element model of polymer anti-seepage dam wall considering the coupling effect of seepage field and stress field was established. Compared with the 2D model, the 3D model is more similar to the actual situation of the dam, and the 3D model can better reflect the complex working conditions, especially those applied to the top of the dam, such as traffic loads. Based on the 3D finite element model, the influence of axle load, vehicle speed and driving position on the stress-deformation characteristics of the dam and polymer anti-seepage wall was analyzed, with the displacement difference of the dam and

anti-seepage wall, the Mises stress of the wall and the subsidence of the wall as indicators. The main conclusions are as follows:

Under the traffic load and pore pressure, the response of the middle dam is larger, but the top of the dam is more sensitive to the traffic load, and the influence depth of the traffic load on the dam strain is about 5 m.

Compared with a traditional concrete wall, the capacity of coordinative deformation between the polymer anti-seepage wall and dam is better, and the ratio of the displacement difference between the polymer and rigid concrete at the dam crest with the dam is about 1:96.

The subsidence and Mises stress of the dam are positively correlated with traffic load amplitude, and negatively correlated with vehicle speed. The displacement difference between the wall and the dam rises with the increase of vehicle speed and amplitude. When the amplitude rises from 0.7 to 1.5 MPa, the displacement difference increases by 87.1%. Additionally, when the vehicle speed grows from 20 to 100 km/h, the displacement difference increases by 17.8%.

When the driving position approaches the wall, the displacement difference between the wall and the dam gradually decreases, and the Mises stress of the wall first increases slowly and then decreases sharply, both of which reach their minimum values when the vehicle drives just above the anti-seepage wall.

The most influential factors of vehicle load on the coordination characteristics of dam deformation include the size of vehicle axle load and the driving position, while the vehicle speed has little effect on the coordination characteristics of dam deformation.

From the perspective of the deformation coordination between the polymer anti-seepage wall and the dam, a vehicle with low axle load passes directly above the anti-seepage wall as quickly as possible, which exerts less influence on the dam.

**Author Contributions:** Conceptualization, H.F. and B.X.; Data curation, A.T.; Supervision, H.F., J.G., Y.L. and X.G.; Writing—original draft, H.Z.; Writing—review & editing, B.X. All authors have read and agreed to the published version of the manuscript.

**Funding:** This research was funded by the National Natural Science Foundation of China grant number 52109169, the Natural Science Foundation of Henan grant number 212300410279 and the China Postdoctoral Science Foundation grant number 2021M702951.

**Institutional Review Board Statement:** Not applicable.

**Informed Consent Statement:** Informed consent was obtained from all subjects involved in the study.

**Data Availability Statement:** The data that support the findings of this study are available from the corresponding author upon reasonable request.

**Conflicts of Interest:** The authors declare no conflict of interest.

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
