# Peer review of "Coordination Characteristics Analysis of Deformation between Polymer Anti-Seepage Wall and Earth Dam under Traffic Load"

_water, doi:10.3390/w14091442_

Round 1
Reviewer 1 Report
Dear authors,
I have a few questions and comments about the paper.
General concept comments:
1) Unfortunately, there are no line numbers in the document, which causes considerable complications in the written communication between the reviewer and the authors.
2) The sentence in Chapter 1 states "Currently, the analysis on traffic load mainly focuses on the field of roads [16-18], while few studies are conducted on the impact of traffic load on dam with polymer anti-seepage wall". Would it be possible to flesh out these studies (provide a reference and describe what results have been achieved)?
3) At the end of results (page 20), I would expect some general summary of the benefits of the 3D model and a more general impact on a other/similar application.
Specific comments:
I) Below Figure 4 si shown "and the driving position is represented by the X coordinates of the central line of the axle". Where the x-coordinate is meant (best linked to another image)?
II) On page 8 it says:
"Fig.7 shows that in the range of 0-4m, the Mises stress gradually increases with the height of the wall. And in the range of 4-15m, it gradually increases with the increase of the wall height. In the range of 15-17m, a certain fluctuation occurs, with the maximum position of the amplitude appearing at the wall height of 15m."
II a) Does Mises stress really increase in the range of 4 - 15 m?
II b) Is it correct that the maximum position of the amplitude occurs at a wall height of 15 m (minimum Miss stress)?
III) In chapter 4.2.3. is written: "This results from the higher pore water pressure on the left side and the coupling of seepage stress".
I was expecting a mention of the water pressure in the pores earlier in the Figures above!
Best reagards
reviewer
Reviewer 2 Report
After a full review for this manuscript, there are several issues required to be improved by the authors.
1. The draft stated that it has considered the coupling effect of seepage field and stress field. But these aspects are not found throughout the text, and the mechanism of coupling effect are not discussed in detail. Moreover, the parameters of material permeability are not given. Therefore, these details cannot support the results of this draft.
2. In general, for the dam top surface with traffic requirements, the upstream slope surface of the dam are strictly treated by impermeable anti-seepage materials such as geomembrane, instead of the case used in the 3D finite element model. So, the rationality of the analyzed model should be reconsidered according to the actual engineering conditions.
3. As shown in the results (e.g., Figs. 9 and 13), the magnitude of displacement different and subsidence calculated are very low relative to the real situations. In other words, the errors of these quantities are allowable when making the engineering design. Therefore, although the results reveal the basic variation trend of the anti-seepage wall and the dam under traffic load, but they contribute little to the risk assessment of the dam operation.
The above issues need to be reconsidered reasonably. After a major revision, it can be considered for possible publication in this journal.
Reviewer 3 Report
In this paper, the coordination characteristics of the polymer anti-seepage wall and the deformation of the earth dam under the action of traffic load are analyzed, and a three-dimensional finite element model of the earth dam considering the coupling effect of the seepage field and the stress field is established. The current topic is of great interest, the analyses are appropriate, and some significant findings are found. Hoping to improve the quality of the manuscript, some brief suggestions are provided as follows.
- The introduction of the model size is incomplete, it is recommended to add the dimensions of the crest width and slope ratio. And it is recommended to illustrate the settings of different driving positions. And it is recommended to use a schematic diagram to introduce the settings for different driving positions.
- In the analysis of Figure 7, the analysis shows that "in the range of 4-15m, it gradually increases with the increase of the wall height." This does not match the changing trend of the curve in the figure. Please check it.
- Is there any corresponding engineering case which can verify the correctness of model in the paper? If there is, add it into the paper.
- The research background analysis is not enough. Some of the latest research results should be added.
Round 2
Reviewer 2 Report
The permeability of dam foundation is assumed to be uniform, which is not always true in reality (e.g. Direct simulation methods for a class of normal and lognormal random fields with applications in modeling material properties). The literature review is suggested cover this part.
